# The Effect of *p*-Coumaric Acid on Browning Inhibition in Potato Polyphenol Oxidase-Catalyzed Reaction Mixtures

**DOI:** 10.3390/foods11040577

**Published:** 2022-02-17

**Authors:** Shu Jiang, Michael H. Penner

**Affiliations:** 1College of Food and Pharmaceutical Sciences, Ningbo University, Ningbo 315800, China; 2Department of Food Science and Technology, Oregon State University, Corvallis, OR 97331-6602, USA

**Keywords:** enzymatic browning, inhibition, *p*-coumaric acid, model system, potato juice

## Abstract

There has been considerable interest in using natural polyphenol oxidase (PPO) inhibitors to control browning in fruit and vegetable products. *p*-Coumaric acid (*p*CA), a common secondary metabolite of plants, has been studied as an inhibitor of PPOs/tyrosinases from several foods (e.g., mushroom, apple, and potato). However, studies on the use of *p*CA for the inhibition of PPO-initiated browning in actual food systems are limited. Therefore, a study was carried out to ascertain the efficacy of using *p*CA to limit PPO-initiated browning in fresh potato juice. The extent of browning inhibition by *p*CA was shown to be reaction system-dependent. Browning in potato juice was unexpectedly enhanced by the addition of *p*CA. This was interpreted as *p*CA acting as an alternative substrate with significantly higher browning efficiency; extent of browning under this condition was higher than that observed in the native potato juice. The addition of *p*CA to any of the model reaction mixtures (i.e., those containing semi-purified enzymes and substrates) significantly inhibited browning. The discrepancy in *p*CA effects on browning inhibition in different reaction systems is postulated to be mainly due to non-enzyme and non-substrate components in potato juice that participate in the post-PPO reaction sequences, which ultimately lead to brown color formation.

## 1. Introduction

Enzymatic browning leading to unwanted tissue discoloration is a significant concern in the food industry because it negatively affects food quality (particularly appearance) and thus lowers consumer acceptability. Thus, unwanted enzymatic browning results in large amounts of food waste. This browning (production of melanin pigments) is a consequence of a series of reactions initiated by the polyphenol oxidase (PPO)-catalyzed phenolic compound oxidation to quinones [1]. The PPO reaction is common to raw, oxygen-exposed, plant-based foods that have suffered cell disruption, particularly during fruit and vegetable postharvest handling, storage, and processing. Thus, there remains considerable interest in developing strategies to control browning in foods. Enzymatic browning is commonly delayed using one or more of three different approaches: (1) enzyme inhibition/inactivation, (2) quinone reduction/modification, and (3) substrate (i.e., phenolics, oxygen) removal/modification [2]. Chemical anti-browning agents that have been studied for these purposes include ascorbic acid and its derivatives [3], citric acid [4], sulfites, sulfur-containing organics (e.g., cysteine, glutathione) [5], ethylenediaminetetraacetic acid (EDTA) [6], 4-hexylresorcinol, kojic acid, aromatic carboxylic acids (e.g., benzoic and cinnamic acids), halides (e.g., sodium chloride), and cyclodextrins [1,7]. Sulfites were among the most widely used anti-browning agents for fresh fruit and vegetable products, but their use was banned by the FDA in 1986 due to health concerns. Nowadays, ascorbic acid is probably the most widely used anti-browning agent in the food industry. Unfortunately, ascorbic acid works by reducing quinones back to the corresponding di-phenols and thus it only temporarily controls browning [2]. Once the ascorbic acid in a system is consumed, then quinones will build up and brown color will form. Hence, alternative anti-browning agents need to be investigated.

There is considerable interest in searching for natural PPO inhibitors to control browning in fruit and vegetable products. *p*-Coumaric acid (*p*CA), a common secondary metabolite obtained from plants, has been studied as an inhibitor of PPOs/tyrosinases from several foods (e.g., mushroom, apple, and potato) [8,9,10]. *p*CA inhibition of PPO is reported to be reversible due to its competitive or mixed-type (non-competitive in some cases) inhibitory behavior under different reaction conditions depending on substrates and enzymes from various sources [8,9,11,12,13]. The interaction between *p*CA and PPO in the presence of a natural substrate, such as tyrosine, is commonly explained by *p*CA competing with tyrosine for the active site of the enzyme due to its structure being similar to that of tyrosine (see Figure 1) [14]. In fact, *p*CA can be oxidized by PPO from several sources (e.g., mushroom and potato) [8,15], confirming that it can reside in the active site. Thus, being an alternative substrate for PPO, *p*CA appears to inhibit PPO-catalyzed oxidation of the natural phenolics endogenous to fruits and vegetables, such as tyrosine. The oxidation of *p*CA results in the formation of a different quinone (i.e., *o*-caffeoquinone), which leads to different quinone-derived products. The result is that *p*CA inhibits the formation of the melanin pigments that would naturally be derived as a result of tyrosine oxidation [15,16].

The *p*CA effect on inhibiting PPO-catalyzed reactions is well studied in model systems comprised of enzymes and substrates in buffer solutions [8,9,13]. However, studies on the use of *p*CA for the inhibition of PPO-initiated browning in actual food systems are limited. Anti-browning effects were found in actual food systems (e.g., potato/apple puree and fresh-cut banana slices) when *p*CA-containing plant extracts, such as rice bran extract and pineapple shell extract were used [17,18]; the anti-browning nature of these extracts was attributed to the *p*CA within them. The individual effect of *p*CA on delaying browning was only found in mushroom from a single study [19]. Thus, the direct/individual effect of *p*CA on browning inhibition in actual food systems has not been extensively investigated. This is certainly relevant to understanding the nature and application of *p*CA for the control of enzymatic browning in foods. Arguments in favor of investigating *p*CA as an anti-browning agent for use in fruits and vegetables include: (1) *p*CA has been shown to inhibit PPO-catalyzed reactions and related browning in model systems; (2) *p*CA is widely distributed in plants and thus potentially readily available (e.g., rice/corn bran, pineapple shell, and ginseng leaves) [12,17,18,20]; and (3) *p*CA is a natural compound that is likely to be ‘safe’ for food applications.

The present study uses freshly-prepared potato juice as a common food browning system to investigate the effect of *p*CA on browning inhibition. Potato juices readily undergo enzymatic browning [21]; the browning associated with potato juice occurs in stages, from yellow to pink to brown to black. Severe blackening of potato tissues and juices has been attributed to the oxidation of the endogenous substrate, tyrosine [22]. This discoloration undoubtedly lowers the quality of potato products (color is often considered as the first quality parameter evaluated by consumers [23]). There is limited information related to the color development in potato juice systems associating with substrates/inhibitors. Thus, studying such a system would provide more information about color development/reduction. Although fresh potato juice is not commercially produced for consumers, it is used as the model food system because understanding the color development in potato juice will help us understand the process in potato tissue (e.g., fresh-cut potatoes) and the juice is far easier to use experimentally.

The aim of this study was to investigate the effect of using *p*CA on browning inhibition in a potato juice system. A relatively defined model system is included for comparative purposes. The specific questions addressed in this study are: (1) Does *p*CA inhibit browning in fresh potato juice? (2) Does *p*CA behave the same towards browning inhibition in a model and fresh potato juice system?

## 2. Materials and Methods

### 2.1. Materials

Russet potatoes were purchased from local markets. *p*-Coumaric acid (*p*CA) (≥98.0%, HPLC), L-tyrosine (reagent grade, ≥98.0%, HPLC), β-cyclodextrin (βCyD) (≥97%), chlorogenic acid (CA; preferred IUPAC numbering 5-*O*-caffeoylquinic acid; ≥95%), L-ascorbic acid (AA) (reagent grade, crystalline), and tropolone (98%) were purchased from Sigma–Aldrich (USA). Chlorogenic acid stock solutions were prepared in dilute acid (10 mM phosphoric acid) to prevent autoxidation [24].

### 2.2. Preparation of Fresh Potato Juice

Approximately 50 g, weighed to the nearest 0.1 g of peeled, washed, then diced potato (chilled at 4 °C) was mixed in a pre-chilled blender with an equal weight of an aqueous solution (chilled at 4 °C) that was 10 mM AA, 50 mM sodium phosphate, pH 7.0, and homogenized for 30 s. The resulting homogenate was filtered through four layers of cheesecloth followed by centrifugation of the filtrate at 10,000 rpm for 15 min at 4 °C. The collected supernatant, hereafter referred to as fresh potato juice, was kept in an ice bath until it was used for enzymatic browning studies (within 1 h).

### 2.3. Preparation of Semi-Purified PPO

Semi-purified PPO was prepared as described in our previous study [7]. Briefly, a PPO extract (i.e., fresh potato juice) was obtained as described above (see the Section 2.2) with the exception that the homogenizing phosphate buffer contained 30 mM AA. Ice-cold acetone (final concentration of 50%) was added dropwise to the PPO extract to precipitate PPO at −20 °C for 1 h. The resulting suspension was then centrifuged at 10,000 rpm for 30 min at 4 °C. The supernatant was discarded, and the pellet was washed three times with ice-cold acetone. The resulting washed precipitate was dried in an ice bath (in the hood) for approximately 3 h. The obtained dried powder is hereafter referred to as PPO-acetone powder (PPOA). PPOA was then added to 50 mM sodium phosphate buffer, pH 7.0, to give 25 mg PPOA per mL buffer. This suspension was allowed to dissolve for approximately 1 h at 0 °C and then centrifuged at 10,000 rpm for 15 min at 4 °C. The resulting enzyme-containing supernatant, hereafter referred to as semi-purified PPO preparation (SPPO), was decanted and kept in an ice bath until used.

### 2.4. Quantification of Color Formation

Color was determined spectrophotometrically and/or colorimetrically at ambient temperature (~22 °C). Spectrophotometrically, color formation was characterized as an increase in absorbance at 490 nm using a Shimadzu UV160U spectrophotometer with 1-cm disposable cuvettes. Colorimetrically, color formation was also characterized using a Hunter Lab Color Quest colorimeter with a D65 light source and an observer angle at 10° in a transmittance mode. An optically-clear glass cell with 10 mm path length was used. Measured CIE coordinates, *L** (lightness–darkness), *a** (red/green), and *b** (yellow/blue) were used for color evaluation. The *L** parameter was used as a measure of ‘lightness’, the *a** parameter was used as a measure of the relative red–green intensity, and the *b** parameter was used as a measure of relative yellow–blue intensity. Based on these three CIE coordinates, the browning index (BI) was calculated as (1) and (2) [25]:(1)$$\mathrm{BI}=\frac{100\left(x-0.31\right)}{0.172}$$
(2)$$x=\frac{a^{*}+1.75L^{*}}{5.645L^{*}+a^{*}-3.012b^{*}\;}$$

### 2.5. Evaluation of Tyrosine Effect on Color Development in Model Potato Juice

The model potato juice system contained 0.3 mM CA, various concentrations of tyrosine (0–2 mM), and semi-purified enzyme in a 50 mM sodium phosphate buffer solution at pH 7.0. Enzyme reactions were initiated by adding 0.1 mL SPPO to 2.1 mL of a temperature-equilibrated substrate solution. After a two-hour reaction at ambient temperature (approximately 22 °C), color spectra were then measured spectrophotometrically with a wavelength range of 400–800 nm.

### 2.6. Determination of pCA Concentration Effect on Color Formation in a Model and Fresh Potato Juice System

In the model system, reaction mixtures contained 0.05 mM CA, 2 mM tyrosine, and various concentrations of *p*CA (0–5 mM) in a 50 mM sodium phosphate buffer solution at pH 7.0. Enzyme reactions were initiated by adding 0.1 mL SPPO to 2.1 mL above the reaction mixtures. After a two-hour reaction at ambient temperature (approximately 22 °C), color was measured spectrophotometrically as described above. In a fresh potato juice system, enzyme reactions were initiated by adding 0.4 mL potato juice to 2 mL of 50 mM phosphate buffer solution at pH 7.0, containing *p*CA amounts such that final reaction mixture concentrations of *p*CA ranged from 0–5 mM. After a two-hour reaction at ambient temperature (approximately 22 °C), color was measured spectrophotometrically as described above.

### 2.7. Determination of pCA and/or βCyD Effect on Color Formation in a Model and Fresh Potato Juice System

In the model system, reaction mixtures contained 2 mM tyrosine, 0.05 or 0.3 mM CA, 5 mM *p*CA, or 10 mM βCyD, or a combination of the latter two in a 50 mM sodium phosphate buffer solution at pH 7.0. Enzyme reactions were initiated by adding 0.1 mL SPPO to 2.1 mL above the reaction mixtures. PPO reaction mixtures were incubated for 2 h. Color was then measured spectrophotometrically and colorimetrically as described above. In the fresh potato juice system, reaction mixtures contained 5 mM *p*CA, or 10 mM βCyD, or a combination of the latter two in a 50 mM sodium phosphate buffer solution at pH 7.0. Enzyme reactions were initiated by adding 0.4 mL fresh potato juice to 3 mL above reaction mixtures. PPO reaction mixtures were incubated for 2 h. Color was measured spectrophotometrically as described above.

### 2.8. Effect of Juice Concentration on pCA and/or βCyD Inhibition of Color Formation

Reaction mixtures contained 5 mM *p*CA, or 10 mM βCyD, or a combination of the latter two in a 50 mM sodium phosphate buffer solution at pH 7.0. Enzyme reactions were initiated by adding 0.2, 1.0, or 2.0 mL fresh potato juice to 8.6, 7.8, or 6.8 mL above the reaction mixtures, respectively. Color was measured spectrophotometrically as described above.

### 2.9. Effect of Tropolone on Color Formation in pCA-Containing Fresh Potato Juice

Reaction mixtures contained 0.2 mM tropolone and various concentration of *p*CA (0–5 mM) in a 50 mM sodium phosphate buffer solution at pH 7.0. Enzyme reactions were initiated by adding 0.4 mL fresh potato juice to 2 mL above reaction mixtures. After a two-hour reaction at ambient temperature (approximately 22 °C), color was measured spectrophotometrically as described above. Control experiments were treated identically but without tropolone.

### 2.10. Statistical Analysis

The data were reported as means ± standard deviation of triplicate experiments. Statistical differences between different treatments (groups) were determined by analysis of variance (ANOVA) and Tukey’s post-hoc test (*p* ≤ 0.05). Statistical analyses were performed using Minitab 16 (Minitab LLC. State College, PA, USA) software.

## 3. Results and Discussion

Initial experiments were aimed at characterizing color development in the model system, which is based on potato PPO and substrates endogenous to the potato. The goal of these initial experiments was to have a well characterized, relatively simple system against which to compare the character of color development in the actual potato juice system. The effect of tyrosine on color development in the SPPO/CA model system was first evaluated (Figure 2). CA and tyrosine are the primary endogenous PPO substrates of potato. Ratios of tyrosine to CA for this experiment were 0, 3.33, and 6.67, simulating the ratio of these substrates in potato tubers [26,27,28]. Comparing with the color spectrum that is solely due to PPO/CA reactions, PPO reactions with a mixture of tyrosine and CA clearly generated a pink/reddish product, which was maximumly absorbed at approximately 490 nm. The compound formed was likely to be dopachrome, produced from auto-oxidation of dopaquinone, an initial product from PPO-catalyzed tyrosine/L-DOPA oxidations [16]. At physiological pH, unstable dopachrome undergoes rearrangement forming intermediates that can react with other compounds, especially with nucleophiles, resulting in black/dark brown colored products [29]. Such a color change (from pink to black) in compounds was actually observed in a reaction mixture containing SPPO and tyrosine during a long reaction time (about 24 h) at room temperature. However, when PPO reacted with CA under the same conditions, the color of reaction mixtures changed from yellow to brown instead. It has been reported that quinones generated by PPO/CA reactions are yellowish/greenish and subsequent quinone reactions with other compounds (e.g., amino acids and proteins) lead to polymerization of brown pigments [2]. This indicates that tyrosine is the substrate, not CA, that contributes the color formation from pink to black in the model system, which is actually very similar to that in a fresh potato juice system. In fact, it has been reported that tyrosine is associated with blackspot formation in potato tubers [30]. Thus, with tyrosine being a more significant substrate in color development in potato, the pink/reddish color was particularly studied (i.e., color absorbance measured at 490 nm) in the potato juice system.

The primary question addressed in this study is the effect of *p*CA on color formation in an actual potato juice system. In combination with *p*CA, β-cyclodextrin (βCyD) was also included in this study to determine the potential of using it for browning inhibition in present experimental systems, as βCyD is considered as a natural anti-browning agent in foods and has been shown to slow browning in fruit and vegetable juices. For simplicity, the *p*CA and/or βCyD effect on PPO-initiated color formation was first evaluated in the model system and then the fresh potato juice system. With respect to the model system, after 2-h PPO reactions with tyrosine and CA in the absence of *p*CA and/or βCyD, it was found that the higher the CA concentration was, the higher the absorbance of the reaction mixture was (Figure 3). This indicates enzyme affinity to different substrates. In the case of current systems, a smaller K_m_ value of potato PPO for CA (0.14 mM) comparing with that for tyrosine (1.4 mM) indicates CA is a preferable substrate for potato PPO [28]. In addition, PPO enzymes generally oxidize diphenols more rapidly than monophenols [6]. Thus, CA is the substrate that is rapidly oxidized by PPO. Being a monophenol, tyrosine is very slowly oxidized by potato PPO [8]. Indeed, with higher concentration of CA (0.3 mM) in the reaction mixture, the color first observed (about 1 min after reaction initiation) was quite yellow/light green, which is likely to be quinone products generated from CA oxidation [2]. The color then changed to pink/orange after 2-h reactions. As has already been pointed out, the pink/reddish color is more likely due to dopachrome, coming from PPO/tyrosine oxidation. In the presence of a low concentration of CA (0.05 mM), the yellow color due to CA oxidation was not visually observed but the pink color was shown instead. This indicates that in the case of the model system, PPO oxidizes CA faster than tyrosine, and the ratio of tyrosine and CA decides the relative intensity of pink and yellow color.

In the presence of 5 mM *p*CA, the color formation was significantly reduced (Figure 3) compared with the control system (no *p*CA added). This demonstrates an inhibition of color formation in the model system by *p*CA. *p*CA has been reported as a competitive inhibitor of potato PPO when monophenol *p*-cresol is a substrate, and a mixed-type inhibitor when CA is a substrate [8]. With respect to the current system, since *p*CA can be very slowly oxidized by potato PPO [8], *p*CA could act as an alternative substrate that possibly competes with tyrosine used by PPO, resulting in a decrease in dopachrome initiated by tyrosine oxidation. Note that color inhibition by *p*CA would only occur when the product generated from PPO/*p*CA reaction, and its secondary reactions tends to show less color compared with color from dopachrome due to tyrosine oxidation. In the presence of 10 mM βCyD, the color formation was significantly reduced compared with when βCyD was absent, but the extent of color reduction was less than when 5 mM *p*CA was present (Figure 3). The mechanism of βCyD inhibiting potato PPO-initiated color formation is that βCyD forms an inclusion complex with CA resulting in a reduction of free CA concentration and slower PPO/reaction with CA. With respect to PPO/tyrosine reactions, βCyD does not have a significant effect due to its low binding capacity with tyrosine. Our previous work has already shown βCyD does not directly inactivate PPO in potato [7]. 

When both *p*CA and βCyD were included in the reaction mixture, the extent of color inhibition was higher than when either *p*CA or βCyD was used alone, especially in the presence of high CA concentration (Figure 3). This also indicates a strong interaction between βCyD and CA but not *p*CA. It has been reported that in water, the βCyD/*p*CA binding constant (stability constant) is 160 M^−1^ [31], which is approximately four times smaller than βCyD/CA (646 M^−1^) [32]. Thus, βCyD does not bind *p*CA as strongly as CA. The effect of *p*CA on color formation would not be limited as the amount of free *p*CA that can interact with PPO does not significantly change in the presence of βCyD. A combined use of βCyD and *p*CA does not decrease the extent of color inhibition but even slightly increases such color reduction (see Figure 3).

To better characterize the color formation in the model system, color was also measured colorimetrically. The result shows *L** values in the presence of pCA and/or βCyD are higher than the control (with pCA and/or βCyD absent), especially in the presence of higher concentration of CA (0.3 mM) (Figure 4a). This indicates the addition of pCA and/or βCyD lightens the color of reaction mixtures, with a combined use of pCA and βCyD being the most efficient on color inhibition. However, all lightness values were very high (> 90) due to the model system itself and also due to the path length of colorimeter transmittance cell that was used. In the presence of pCA and/or βCyD, both *a** and *b** values decreased compared with the absence of pCA and/or βCyD (Figure 4b,c). This indicates both red and yellow color were significantly reduced by pCA and/or βCyD. Figure 4d compares the browning index (BI) in the presence of pCA and/or βCyD; browning (i.e., color formation) in such a model system was significantly inhibited by pCA and/or βCyD, with the inhibitory effect of a combined use of pCA and βCyD being the greatest. Based on color parameters (*L**, *a** and *b**) and the BI, pCA alone has a higher inhibitory effect on color formation than βCyD alone. However, the addition of βCyD enhanced the inhibitory effect of pCA on color formation in the model system. The corresponding pictures that better demonstrate the effect of pCA and/or βCyD on browning formation in the model system are shown in Figure 4e.

Data presented in the model system show *p*CA significantly inhibited color formation, with an enhanced inhibitory effect in the presence of βCyD. However, in a fresh potato juice system, the addition of the same amount of *p*CA enhanced color formation, with a lesser extent of color enhancement in the presence of βCyD compared with fresh potato juice containing no *p*CA and βCyD (Figure 5). Further experiments evaluated the effect of *p*CA concentration on color formation in both model and fresh potato juice systems. The results show color formation was reduced, depicted as decreases in absorbance with increasing *p*CA concentrations in the model system containing substrates and semi-purified enzyme (up to 60% color inhibition at 5 mM *p*CA). However, the addition of *p*CA, especially at concentrations 0.5 and 1.0 mM, significantly increased the color absorbance in fresh potato juice compared with no *p*CA added (Figure 6). This is similar to what was found in Figure 5. This indicates new compounds might be generated in fresh potato juice with the addition of *p*CA. Thus, *p*CA is likely to be an alternative substrate of potato PPO or other enzymes (e.g., peroxidase). Indeed, when potato PPO preparations (SPPO or fresh juice) were incubated with *p*CA only, color slowly formed within 2 h; this effectively demonstrates the oxidation of *p*CA catalyzed by PPO. It is reasonable because *p*CA has been reported to be very slowly oxidized by potato PPO [8]. To test if the color enhancement is due to PPO, a PPO-specific inhibitor, tropolone, was included in the fresh potato juice system. Figure 7 shows that in the presence of tropolone, color formation was completely inhibited either with or without *p*CA. This suggests that PPO, other than other enzymes (e.g., peroxidase), is responsible for color enhancement with *p*CA present, possibly due to *p*CA being a substrate in the fresh potato juice system. Being a substrate of PPO, *p*CA is possibly hydroxylated to caffeic acid by PPO, and the latter is then oxidized to caffeic acid quinone. The similar reactions occur in the spinach–beet PPO system [33]. In this case, the caffeic acid quinone would subsequently react forming dark red/brown pigments, differing from black pigments produced by tyrosine oxidation. An observation regarding this is in the presence of *p*CA; brown pigments were eventually formed (after about 24 h) compared with black pigments produced in the absence of *p*CA. This indicates that by interacting with PPO, *p*CA might alter the final colored product that should be initiated by PPO reactions with endogenous substrates in the fresh potato juice.

When different amounts of fresh potato juice were added to reaction mixtures containing *p*CA and/or βCyD, the effect of *p*CA on color formation was similar. *p*CA significantly enhances browning in fresh potato juice at all concentrations, but a reduced extent of color enhancement was found when the highest amount of potato juice was added (Figure 8). High amounts of fresh potato juice are associated with the more concentrated enzyme, endogenous substrates, and other compounds. A reduced extent of color enhancement is probably due to: (1) endogenous substrates competing with PPO reactions and *p*CA, thus affecting the color formation due to PPO/*p*CA reactions; and (2) other compounds in the potato juice inhibiting the color formation due to PPO/*p*CA reactions.

When comparing the effect of pCA on browning inhibition in the fresh potato juice system with the model system, the effect of pCA on browning inhibition was shown to be reaction-system dependent. The discrepancy in pCA effects on browning inhibition in different reaction systems is possibly due to several reasons: (1) pCA is more rapidly oxidized in the fresh potato system than in the model system, probably due to the low catalytic efficiency of semi-purified PPO on pCA compared with a crude enzyme; and (2) non-enzyme and non-substrate components in fresh potato juice participate in the post-PPO reaction sequences that ultimately lead to enhanced brown color formation.

## 4. Conclusions

The present study evaluated the efficacy of using *p*CA to limit PPO-initiated browning in fresh potato juice. The addition of *p*CA in the fresh potato juice system unexpectedly enhanced browning but significantly inhibited browning in the model system. The extent of browning inhibition by *p*CA appears to be reaction-system dependent. The difference in *p*CA effects on browning inhibition is probably mainly due to endogenous components (e.g., non-enzyme and non-substrate) present in fresh potato juice that can participate in efficient browning formation after PPO-catalyzed reactions. The crude enzyme in fresh potato juice that can oxidize *p*CA more efficiently than a semi-purified PPO preparation may also contribute to such disparity. By working with different reaction systems, the present study demonstrates the difficulty in applying PPO inhibitors to an actual food system, which is more complex and requires various factors to be considered. Further experiments related to potato juice discoloration/color formation progression in the presence of *p*CA and endogenous PPO substrates need to be conducted; this will contribute to a better understanding of the enzymatic browning inhibition in actual food systems.

## Figures and Tables

**Figure 1 foods-11-00577-f001:**
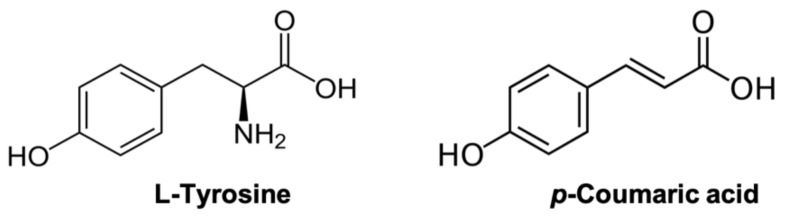
Structure of L-tyrosine and *p*-coumaric acid (*p*CA).

**Figure 2 foods-11-00577-f002:**
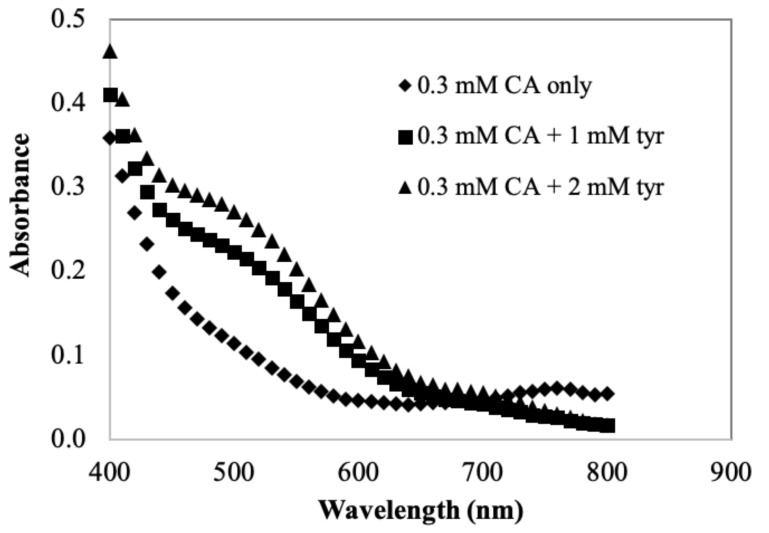
Color spectra of a model system containing semi-purified PPO preparation and substrates. Reaction mixtures were 0.3 mM CA, 0–2 mM tyrosine (tyr), 50 mM sodium phosphate buffer, and pH 7.0. Reactions were initiated by adding 0.1 mL SPPO to 2.1 mL reaction mixture. Spectrums were measured spectrophotometrically after two-hour reactions.

**Figure 3 foods-11-00577-f003:**
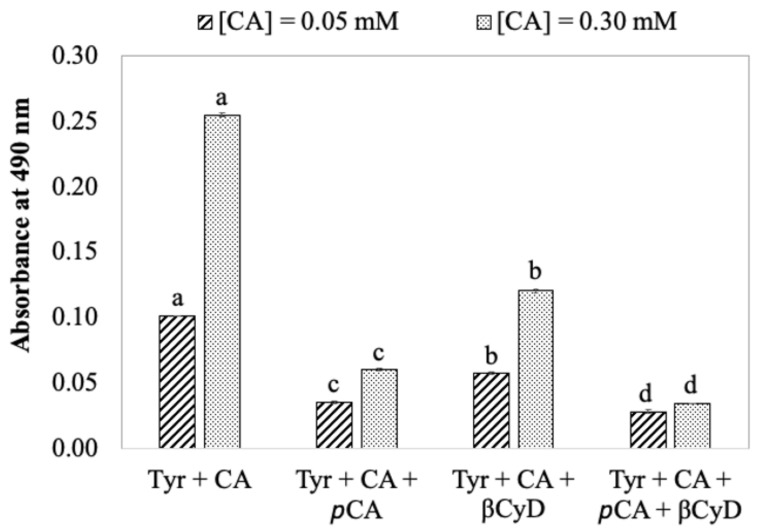
Effect of pCA, βCyD, and a combination of pCA and βCyD on color formation, depicted as an increase in absorbance, in a model system. Reaction mixtures were a potato endogenous substrate solution (2 mM tyrosine (tyr) and 0.05 mM or 0.3 mM CA), and 5 mM pCA or 10 mM βCyD, or a combination of pCA and βCyD. Enzyme reactions were initiated by adding 0.1 mL SPPO to 2.1 above the reaction mixture. Color absorbance was spectrophotometrically measured at 490 nm after two hours of PPO reactions at room temperature. Values are means ± standard deviations from triplicate assays. Mean values that do not share a letter are significantly different (*p* ≤ 0.05) at the same CA concentration.

**Figure 4 foods-11-00577-f004:**
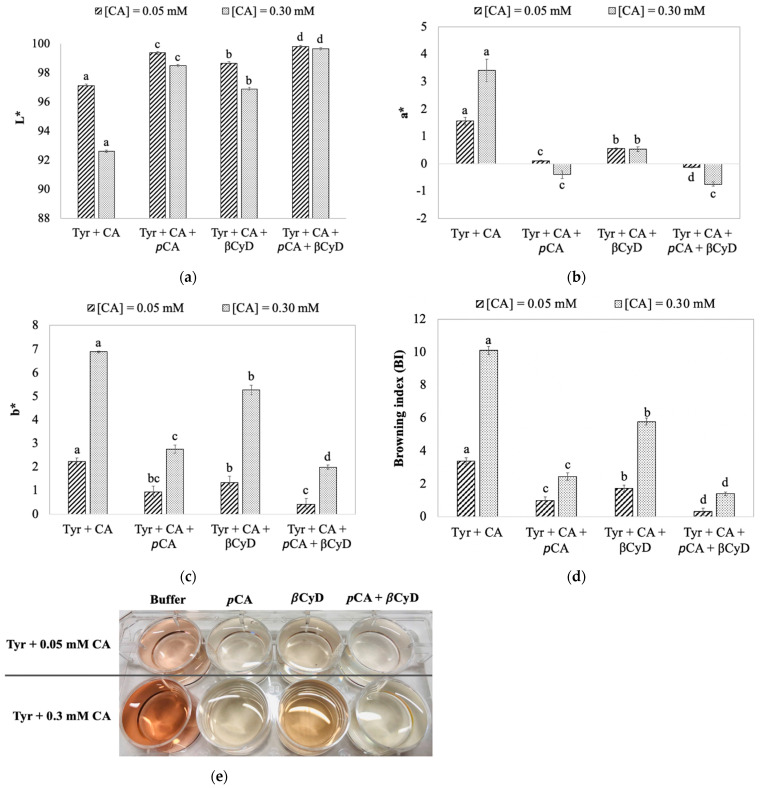
Effect of *p*CA, βCyD, and a combination of *p*CA and βCyD on color formation, depicted as change in color parameters *L** (**a**), *a** (**b**), and *b** (**c**); browning index (BI) (**d**), and color formation (**e**) in the model system. Reaction mixtures are as in Figure 3. Enzyme reactions were initiated by adding 0.6 mL SPPO to 12.6 mL reaction mixture. Color was colorimetrically measured after two-hour PPO reactions at room temperature. Values are means ± standard deviations from triplicate assays. Mean values that do not share a letter are significantly different (*p* ≤ 0.05) at the same CA concentration.

**Figure 5 foods-11-00577-f005:**
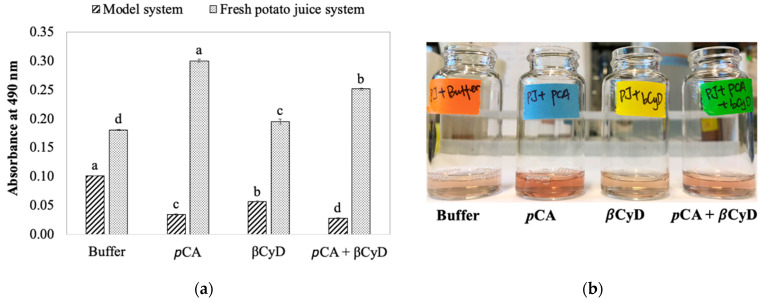
(**a**) Comparison of pCA and/or βCyD effects on color formation in a model and fresh potato juice (PJ) system. In a model system, reaction mixtures contained 2 mM tyrosine, 0.05 mM CA, and 5 mM pCA or 10 mM βCyD, or a combination of pCA and βCyD. Enzyme reactions were initiated by adding 0.1 mL SPPO to 2.1 above the reaction mixture. In a fresh potato juice system, reaction mixtures contained 5 mM pCA, or 10 mM βCyD, or a combination of the latter two in 50 mM sodium phosphate buffer, pH 7.0. Enzyme reactions were initiated by adding 1 mL fresh potato juice to 7.8 mL above reaction mixtures. (**b**) Observation of color formation in fresh potato juice with and without pCA and/or βCyD. Values are means ± standard deviations from triplicate assays. Mean values that do not share a letter are significantly different (*p* ≤ 0.05) within the same system.

**Figure 6 foods-11-00577-f006:**
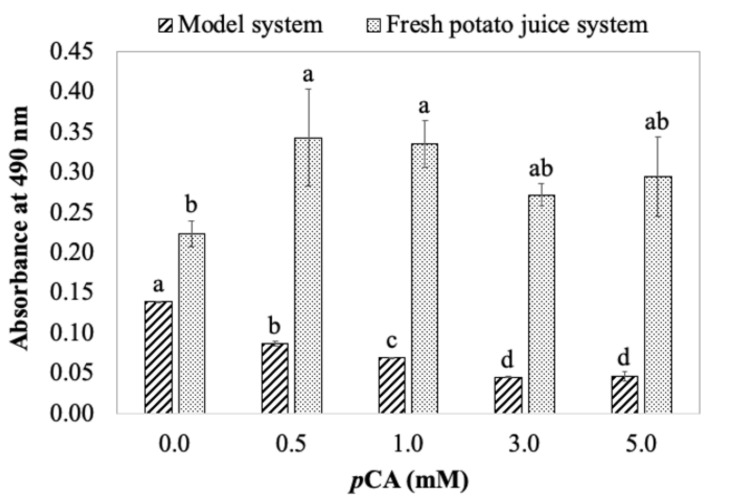
Effect of pCA concentration on color formation, depicted as increases in absorbance, in a model and fresh potato juice system. In the model system, reaction mixtures contained 0.05 mM CA, 2 mM tyrosine, and various concentration of pCA (0–5 mM) in 50 mM sodium phosphate buffer, pH 7.0. Enzyme reactions were initiated by adding 0.1 mL SPPO to 2.1 mL above reaction mixtures. In a fresh potato juice system, reaction mixtures contained various concentrations of pCA (0–5 mM) in 50 mM sodium phosphate buffer, pH 7.0. Enzyme reactions were initiated by adding 0.4 mL fresh potato juice to 2 mL above reaction mixtures. Values are means ± standard deviations from triplicate assays. Mean values that do not share a letter are significantly different (*p* ≤ 0.05) within the same system.

**Figure 7 foods-11-00577-f007:**
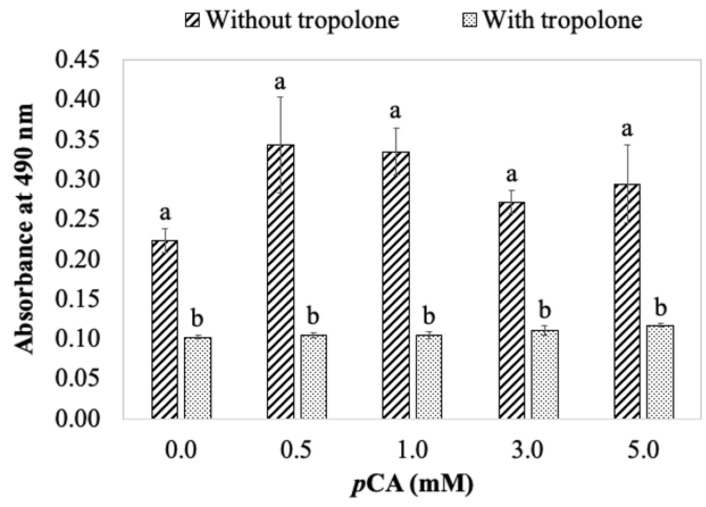
Effect of tropolone on color formation depicted as increase in absorbance, in a fresh potato juice system. Reaction mixtures contained 0.2 mM tropolone and various concentrations of pCA (0–5 mM) in 50 mM sodium phosphate buffer, pH 7.0. Enzyme reactions were initiated by adding 0.4 mL fresh potato juice to 2 mL above reaction mixtures. Control experiments were treated identically but without tropolone. Data points represent means ± standard deviation from triplicate assays. Mean values that do not share a letter are significantly different (*p* ≤ 0.05) at the same pCA concentration.

**Figure 8 foods-11-00577-f008:**
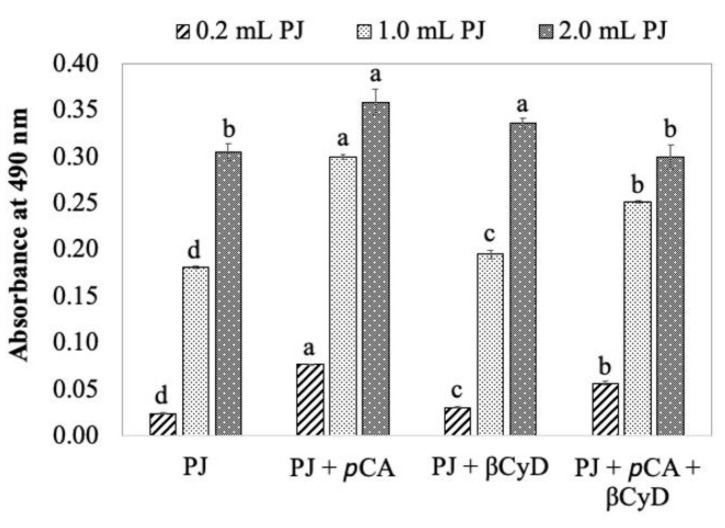
Comparison of *p*CA and/or βCyD effects on color formation in reaction mixtures containing various amounts of fresh potato juice (PJ). Reaction mixtures contained 5 mM *p*CA, or 10 mM βCyD, or a combination of the latter two in 50 mM sodium phosphate buffer, pH 7.0. Enzyme reactions were initiated by adding 0.2, 1.0, or 2.0 mL fresh potato juice to 8.6, 7.8, or 6.8 mL above reaction mixtures, respectively. Values are means ± standard deviations from triplicate assays. Mean values followed by different letters are significantly different (*p* ≤ 0.05) at the same concentration of fresh potato juice.

## Data Availability

Data is contained within the article.

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
