# Peer review of "The Effect of p-Coumaric Acid on Browning Inhibition in Potato Polyphenol Oxidase-Catalyzed Reaction Mixtures"

_foods, 2022, doi:10.3390/foods11040577_

Round 1
Reviewer 1 Report
The following study investigates the use of p-coumaric acid (pCA) to inhibit polyphenol oxidase (PPO) mediated browning of potato food products using two assay systems. Findings indicate that in a semi purified PPO extract (SPPO), pCA exhibits some level of competitive inhibition towards the generation of reaction products that absorb light at 490 nm relative to a reaction mixture containing tyrosine (PPO substrate) alone. Inhibition of PPO activity was also observed with beta cyclodextrin (bCyD) was included. Surprisingly, addition of pCA to potato juice (PJ) had the opposite effect of stimulating the production of metabolite(s) that absorb at 490 nm relative to the SPPO extract. This result/reaction could be inhibited through the addition of PPO inhibitor tropolone.
Overall, the manuscript reads very clearly, represents a complete body of work, and the conclusions are congruent with the results presented.
There are some minor typographical issues that could be fixed.
Line 260: “its secondary reactions intends…” should be changed to “its secondary reaction tends…”
Line 277: Consider rewording the double negative: “Does not decrease the color reduction”
Line 325: “These indicates new colored compounds might be…” should be changed to “This indicates new compounds might be…”
Reviewer 2 Report
The arguments that the authors point out in favor of using pCA as an anti-browning agent for use in fruits and vegetables (lines 79-84)
are the same as those already known for other substances that are also very abundant
in the plant kingdom, such as ascorbic acid. Ascorbic acid added to complex food
models reduces the pH to values ​​below 4.5 In the scientific literature, it has been shown that at this acid pH value,
PPO-catalyzed reactions and related browning are inhibited in model systems. Therefore, pCA does not contribute anything when it comes to inhibiting these browning
reactions that ascorbic acid does not contribute.
Reviewer 3 Report
Manuscript ID: foods-1538403
The effect of p-coumaric acid on browning inhibition in potato polyphenol oxidase-catalyzed reaction mixtures
The manuscript describes results of laboratory experiments on the effect of p-coumaric acid on browning inhibition in potato polyphenol oxidase-catalyzed reaction. The subject is interesting and results can provide an advance in current knowledge to using natural chemicals to inhibit fresh potato browning.
Keywords should not be the repetition of the title words (polyphenol oxidase, browning inhibition, p-coumaric acid). Please find such words which are not be repetition of the title, this way search engines of the web will find your paper with higher probability.
The methods should be more detail described. How many experiments have been done (line 202 – the first experiment, line 318 – further experiments)? What were the experimental factors?
The study results were analyzed statistically using an analysis of variance (ANOVA). What ANOVA model was used?
I suggest to remove lines 192-201 or move to the Introduction.
Lines 267-268 – reference should be added.
Lines 276-278 – Please see Figure 3.
Recommendations for the future experiments which can be based on your study results should be added in the Conclusions.
References should be according to guidelines for Foods. Check the guidelines and make appropriate changes.
